# #LetsUnlitterUK: A demonstration and evaluation of the Behavior Change Wheel methodology

**Julia Kolodko**[1☉], **Kelly Ann Schmidtke**[2☉]*, **Daniel Read**[1☉], **Ivo Vlaev**[1☉]

**1** Warwick Business School, The University of Warwick Coventry, Coventry, United Kingdom, **2** Warwick Medical School, The University of Warwick Coventry, Coventry, United Kingdom

☉ These authors contributed equally to this work.
* Kelly.A.Schmidtke@warwick.ac.uk

**Data Availability Statement:** All data available are available at the Figshare repository at figshare.com/articles/dataset/_LetsUnlitterUK_A_demonstration_and_evaluation_of_the_Behavior_Change_Wheel_Methodology/16895335.

## Abstract

The Behavior Change Wheel is the most comprehensive and practically useful methodology available for developing behavior change interventions. The current article demonstrates how it can be applied to optimize pro-environmental behaviors and, in so doing, give interventionists access to a rigorous set of theories and techniques for systematically developing pro-environmental interventions. Section 1 describes the development of an intervention to increase people's intentions to post anti-littering messages on social media. Study 2 describes the development and evaluation of an intervention to increase people's actual anti-littering posts. Both evaluations are randomized controlled trials that compare the effectiveness of the developed intervention with interventions less informed by the Wheel. We found interventions completely informed by the Wheel to be more effective than interventions less (or not at all) informed by the Wheel. The discussion explores how the Behavior Change Wheel methodology can be used to design future pro-environment interventions.

## Introduction

The Behavior Change Wheel synthesizes insights from 19 domain-specific frameworks and claims to be the most comprehensive and practically useful behavior change framework available [1]. Individually, none of the previous 19 frameworks comprehensively account for sub-optimal behavior nor do they provide a systematic method for intervention design. For example, the Department for Environment, Food, and Rural Affairs' 4-E model describes four reasons pro-environmental interventions may be ineffective but does not help identify which reason(s) are most important to address [2]. Looking at another example, the MINDSPACE framework provides a checklist of nine tools policymakers can use to change behavior by influencing more automatic psychological processes, but it does not provide tools to address more reflective psychological processes nor guidance as to when each tool should be used in particular circumstances [3]. The Behavior Change Wheel seeks to provide a comprehensive and pragmatic eight-step methodology to change behavior.

**Funding:** This project was supported by the National Institute for Health Research (NIHR) Applied Research Centre West Midlands (ARC WM; NIHR200165). The views expressed are those of the author(s) and not necessarily those of the NIHR, ARC, or the Department of Health and Social Care. The funders had no role in the design of the study and collection, analysis, and interpretation of data and in writing the manuscript.

**Competing interests:** The authors have declared that no competing interests exist.

While the Wheel methodology is already widely used to develop health-related interventions [4,5], we are the first to jointly describe the development of interventions using the Behavior Change Wheel and the evaluation of those interventions' effectiveness on intended and then actual behavior. The target behavior investigated here, posting ant-littering messages on social media, should be interpreted as a placeholder for which other behaviors could be inserted. Littering is a significant social problem in the UK and has given rise to charities and government initiatives dedicated to reducing littering. Moreover, it is a problem that has a large behavioral component with many potential barriers [6]. For instance, people may lack knowledge about what materials can be recycled, lack bins to dispose of recycling materials, or lack a desire to search for those bins. Where all three barriers exist, a single-component intervention addressing a single barrier may prove insufficient to decrease littering; rather, in such cases the Behavior Change Wheel recommends developing a multi-component intervention addressing all barriers simultaneously. Littering is, therefore, an important domain for studying behavioral interventions.

The aims of the current study are practical: to develop and evaluate interventions that stand to increase pro-environmental posts online in the United Kingdom using the behavior change wheel methodology. We do not assess individual differences or particular mechanisms of action, e.g., to evaluate the effectiveness of components of the interventions. Section 1 describes the development and evaluation of an intervention to increase people's intentions to post anti-littering messages on social media. Section 2 describes the development and evaluation of a behavior change intervention to increase people's actual anti-littering posts. The two studies in combination demonstrate how the Behavior Change Wheel can be used to develop effective behavior change interventions. The discussion illustrates how this methodology could be employed to change other behaviors beyond the health domains where it is most frequently applied.

## Section 1: Increasing intentions to tweet anti-littering messages

The first step of the Behavior Change Wheel is to define the problem in behavioral terms. This step is vital, because how a problem is defined can influence what solutions are generated [7]). To define the problem, one can use root cause analyses, such as a set of interrogative 'why questions' [8,9]. For example, one may ask, "Why is there so much litter?" and respond, "People litter." Second, one may ask, "Why do people litter?" and respond, "People think that most other people litter." Third, one may ask, "Why do people think that most other people litter?" and respond, "Given the amount of litter, littering appears to be a common or acceptable practice," i.e., littering is a social norm.

Social norms are socially determined, implicit or explicit beliefs that people hold about the prevalence or acceptability of behaviors [10,11]. As social norms are greatly influenced by public displays, not private thoughts [12,13], the prevalence of physical litter in the environment may lead people to wrongly believe that littering is normatively accepted. The social norms approach attempts to change undesirable behaviors by correcting such inaccurate beliefs [14], and it has been successfully employed in several domains, e.g., alcohol consumption [15], illegal drug use [16], sunscreen use [17], and energy consumption [18]. However, we did not define the problem simply as people holding inaccurate beliefs, but asked, "Why don't people publicly express their anti-littering sentiments?" One plausible reason is that people are rarely prompted to do so. Therefore, rather than employing a straightforward social norms approach for correcting inaccurate beliefs, we intervened by allowing participants people to publicly express their anti-littering sentiments.

Steps 2 and 3 of the Behavior Change Wheel are to select and specify a target behavior, by saying who should perform what, where, and when. Ultimately, the target behavior was

specified as follows: people recruited over Prolific with a Twitter account and asked to express their intentions to tweet an anti-littering message on Qualtrics should do so after being prompted. For practical reasons, we wanted to evaluate the behavior change intervention's effectiveness online and needed to select a behavior that could be measured online. Initially, we were not certain we could measure actual social media posting behavior and settled on the proxy measure of intentions to post.

Step 4 of the Behavior Change Wheel is to diagnose why the specified target behavior is not already occurring using qualitative, quantitative, or mixed methods [19–21]. In the present study, a quantitative method was used.

## Methods: Diagnostic Survey 1

### Participants

All participants in the surveys/trials described in the current article were recruited via Prolific (www.prolific.ac). Using Prolific's screening feature, participation was limited to residents of the United Kingdom with a Twitter account. For Diagnostic Survey 1, all participants completed the survey on the 14th of December 2016 and received 0.75 GBP. As this survey was largely exploratory, the sample size adheres to Green's liberal rule of thumb for multiple regression analyses, such that at least 50 participants plus 8 times the number of predictor variables take part [22]. In this case, the number of predictor variables was the number of domains, 14, and so we required at least 112 participants; we recruited 225.

### Materials/Procedures

All surveys/trials were approved by the University of Warwick's Humanities and Social Science Research Ethics Committee (ID: 05/15-16) and administered using Qualtrics (2016–2018). In Diagnostic Survey 1, participants first indicated their informed consent to participate and whether they had a Twitter account (Yes or No). Next, participants were told that the survey would contain questions about anti-littering messages. Participants were provided with examples for what types of information anti-littering messages could contain: encouragement for others to not litter or to clean up litter; warnings about the negative consequences of litter; pictures of litter; or pictures of litterers/fly-tippers.

Next, participants indicated how much they agreed with each of 30 statements about posting anti-littering messages, from 1 (strongly disagree) to 7 (strongly agree), in a random order. Some items were reverse worded. The statements were informed by the Theoretical Domains Framework which condenses 112 empirically and theoretically informed behavior change constructs into 14 domains that describe the barriers and facilitators for a target behavior [23–25]. The domains' definitions and associated statements appear in S1 Appendix. In the initial survey, two domains ('Intentions' and 'Optimism') were assessed using one statement, eight domains ('Skills,' 'Behavioral regulation,' 'Social influences,' 'Environmental context and resources,' 'Social/professional role and identity,' 'Beliefs about capabilities,' 'Belief about consequences,' and 'Emotions') with two statements, and four domains ('Knowledge,' 'Memory, attention and decision processes,' 'Goals,' and 'Reinforcement') with three statements.

Lastly, participants answered questions about their gender (Female, Male, or Other/Prefer not to say) and age.

### Analyses

All analyses were conducted using IBM SPSS version 27. For Diagnostic Survey 1, a Cronbach's alpha was calculated for each domain with relevant items reverse scored. Then, one

statement from each domain that initially contained three statements was removed to improve its reliability [26]. As the present survey contained relatively few items, the typical threshold alpha of .70 was reduced to .50 for a domain's composite score to be considered in further analyses; otherwise, the individual items were considered [27]. Next, the mean composite score for each remaining domain was calculated using responses to its retained statement(s). Lastly, a multiple regression analysis was performed with 'Intentions' as the outcome variable and the domains as predictor variables. The level of significance used to assess each domain was not pre-determined.

## Results: Diagnostic Survey 1

While 214 participants completed the survey, 16 said they did not use Twitter and were removed from further analyses. Of the remaining 198, 68 (34.34%) identified as female and 130 (65.66%) as male. The mean age was 30.98 years ($SD = 9.96$). The average participant completed the survey in about eight minutes ($M = 8.69$, $SD = 4.55$).

As the initial Cronbach's alphas were below the threshold of .50 for 'Environmental context and resources' (.12) and the 'Emotions' (.45), their two items were considered separately in the analysis. For the remaining domains, values of Cronbach's alpha ranged from .51 to .81. Table 1 presents each domain's number of statements retained, alpha, and mean composite score.

Next, a linear regression was performed to understand how 'Intentions' was influenced by the remaining domains. The overall model was significant, $F(15, 182) = 41.24$, $p < .001$, $R^2 = 0.88$. Table 2 displays the results of the regression, along with each domain's contributions. The most significant p-values were for the 'Goals' ($b = 0.31$, $SE = 0.09$, $p < .001$) and 'Social/professional role and identity' ($b = 0.27$, $SE = 0.05$, $p = .001$).

## Brief discussion

Moving forward in the intervention development process, we choose to only focus on the two most significant domains: 'Goals' and 'Social/professional role and identity.' More domains

**Table 1. Cronbach's alpha and composite scores for the Theoretical Domains Framework: Diagnostic 1 survey about behavioral intentions.**

| Domain | Number of statements retained | Cronbach's Alpha | Mean Composite Score |
|---|---|---|---|
| Knowledge | 2 | 0.73 | 6.50 |
| Skills | 2 | 0.68 | 6.09 |
| Memory, attention, and decision processes | 2 | 0.65 | 4.29 |
| Behavioral regulation | 2 | 0.51 | 4.58 |
| Social influences | 2 | 0.69 | 4.36 |
| Environmental contexts and resources–item 1 | 1 | n/a | 6.59 |
| Environmental contexts and resources–item 2 | 1 | n/a | 3.17 |
| Social/professional role and identity | 2 | 0.76 | 4.24 |
| Beliefs about capabilities | 2 | 0.64 | 5.23 |
| Optimism | 1 | n/a | 4.87 |
| Intentions | 1 | n/a | 4.05 |
| Goals | 2 | 0.72 | 4.94 |
| Beliefs about consequences | 2 | 0.76 | 4.89 |
| Reinforcement | 2 | 0.60 | 4.92 |
| Emotions–item 1 | 1 | n/a | 4.87 |
| Emotions–item 2 | 1 | n/a | 4.14 |

**Table 2. Multiple regression predicting 'Intentions' composite score from remaining domains: Diagnostic 1 survey about behavioral intentions.**

| | b | SE(b) | p-value | 95% CI | |
|---|---|---|---|---|---|
| (Constant) | -0.12 | 0.88 | 0.89 | -1.86 | 1.61 |
| Knowledge | -0.07 | 0.12 | 0.57 | -0.31 | 0.17 |
| Skills | -0.19 | 0.10 | 0.07 | -0.39 | 0.01 |
| Memory | -0.15 | 0.07 | 0.03 | -0.29 | -0.01 |
| Behavioral Regulation | 0.32 | 0.10 | 0.002 | 0.12 | 0.52 |
| Social influences | 0.17 | 0.09 | 0.07 | -0.01 | 0.36 |
| Environmental contexts…– item 1 | 0.06 | 0.09 | 0.49 | -0.12 | 0.24 |
| Environmental contexts…– item 2 | -0.02 | 0.05 | 0.62 | -0.11 | 0.07 |
| Social and professional role . . . | 0.27 | 0.09 | 0.001 | 0.10 | 0.44 |
| Belief in capabilities | -0.02 | 0.09 | 0.81 | -0.19 | 0.15 |
| Optimism | 0.20 | 0.08 | 0.01 | 0.05 | 0.35 |
| Goals | 0.31 | 0.09 | <0.001 | 0.13 | 0.48 |
| Belief in consequences | -0.05 | 0.11 | 0.67 | -0.26 | 0.17 |
| Reinforce | -0.01 | 0.11 | 0.94 | -0.23 | 0.21 |
| Emotions–item 1 | 0.04 | 0.09 | 0.65 | -0.13 | 0.22 |
| Emotions–item 2 | 0.10 | 0.05 | 0.06 | -0.003 | 0.20 |
| $R^2$ | .88 | | | | |
| F | 41.24* | | | | |

could have been selected, and in some cases would be required to achieve behavior change. For example, increasing national vaccination rates may require offering the vaccination in different potentially appealing locations (the "Environmental context and resources" domain), increasing awareness (the "knowledge" domain) that those opportunities exist, and overcoming negative feelings towards vaccinations (the "emotion" domain) [28]. The aims of the current study are narrower, aiming to influence only people who already have a Twitter account, and testing these two domains in combination presents a unique opportunity to ultimately evaluate intervention completely and partially informed by the Wheel methodology, as will be seen in step 7. But first, we address steps 5 and 6.

Step 5 of the Behavior Change Wheel aims to identify the most appropriate intervention function(s). The nine possible functions describe what the intervention aims to accomplish at a fairly high level of abstraction: 'Education,' 'Persuasion,' 'Incentivization,' 'Coercion,' 'Training,' 'Restriction,' 'Modelling,' 'Enablement, and 'Environmental Restructuring.' The Behavior Change Wheel provides links between the intervention functions and the domains they are best suited to influence (for all links see reference [1]). Six intervention functions are linked to 'Goals' (Education, Persuasion, Incentivization, Coercion, Modeling, and Enablement) and three to 'Social/professional role and identity' (Education, Persuasion, and Modelling).

To narrow down the number of intervention functions, we drew on the APEASE criteria [1]. APEASE stands for 'Acceptability,' 'Practicality,' 'Effectiveness,' 'Affordable,' 'Side-effects/safety,' and 'Equity.' The 'Incentivization' and 'Coercion' functions were judged unacceptable because they would raise ethics concerns. 'Enablement' was not practical, because we had no ability to change Twitter's interface. The remaining three intervention functions are linked to both the 'Goals' and 'Social/professional role and identity' domains,' and were incorporated into the final intervention: 'Education' (by telling people that they can post), 'Persuasion' (by suggesting that people take the opportunity to post), and 'Modelling' (by providing examples of anti-littering messages).

Step 6 of the Behavior Change Wheel is to identify the most appropriate policy category (or categories). The seven policy categories describe the mechanisms through which the intervention functions can be implemented, again at a high level of abstraction, including: 'Communication/Marketing,' 'Guidelines,' 'Fiscal measures,' 'Regulation,' 'Legislation,' 'Environmental/social planning,' and 'Service provision.' The Behavior Change Wheel provides links between the policy categories and the intervention functions they are best suited to facilitate (for all the links see reference [1]). Five intervention functions are linked to 'Education' and 'Persuasion' (Communication/marketing, Guidelines, Regulations, Legislation, and Service provisions), and two are linked to 'Modelling' (Communications/marketing and Service provisions).

We again used the APEASE criteria to reduce the number of policy categories. Most categories were not available to us. For instance, we did not have the authority to issue 'Guidelines,' 'Regulations,' or 'Legislation.' The remaining two policy categories are linked to all identified intervention functions and were incorporated into the final intervention: 'Communication/marketing' (asking participants to post messages) and 'Service provision' (integrating the Prolific, Qualtrics and Twitter interfaces).

Step 7 of the Behavior Change Wheel is to select the most appropriate behavior change technique(s). The Behavior Change Techniques Taxonomy describes 93 theoretically informed and replicable behavior change concepts, e.g., providing 'feedback on behavior' and 'reframing' [29]. The Behavior Change Wheel provides links between specific techniques and the domains they are best suited to influence [1,30]). Using these links, we developed a multi-component intervention to influence the 'Goals' domain by selecting the 'action planning' and 'goal setting (behavior)' techniques. As the 'Social/professional role and identity' domain is not linked to any techniques, we employed two other techniques, introduced in 'Evaluative Trial 1's' methods section, that might influence that domain. Because these techniques were not explicitly linked to the 'Social/professional role and identity' domain, using them was not driven by the Behavior Change Wheel.

Step 8 of the Behavior Change Wheel is to select an appropriate mode of delivery for the intervention. This is the concrete way the intervention will be implemented. Michie et al. [1] provide several examples for communications-based interventions, such as posters and digital media. We adopted three integrated digital platforms: Prolific (where participants could be recruited), Qualtrics (a survey design software), and Twitter (where participants could implement the target behavior).

Table 3 summarizes the eight steps of the Behavior Change Wheel. The middle column describes how we designed an intervention to increase the intention to post anti-littering messages (or tweets). A randomized controlled trial conducted to evaluate that intervention's effectiveness is described next.

## Method: Evaluative Trial 1

### Participants

Participants completed the survey on the 4th or 5th of August 2017 and received 0.75 GBP.

### Survey-Materials/Procedures

Participants first gave their informed consent and then indicated whether they used Twitter (Yes or No), how often they posted (Never, Yearly, Monthly, Weekly, or Daily), and how much they agreed that litter was a problem in the United Kingdom (1 = strongly disagree to 5 = strongly agree). Then they were reminded that littering was a significant issue in the United Kingdom and told they would be asked to write an anti-littering message that they could post on Twitter. Participants were given the same examples for what types of

**Table 3. Behavior Change Wheel steps and actions.**

| Behavior Change Wheel Step | Brief Description of Actions | |
|---|---|---|
| | **Section 1 Intentions to tweet** | **Section 2 Actual tweets** |
| 1 Define the problem | People do not express their anti-litter sentiments publicly. | |
| 2 Select target behavior | Intentions to tweet | Actual tweeting |
| 3 Specify target behavior | [who] Twitter users recruited from Prolific [what] should express their intentions to tweet an anti-littering message [where] in a Qualtrics survey [when] after being prompted. | [who] Twitter users recruited from Prolific [what] should actually tweet an anti-littering message [where] on Twitter [when] within seven days. |
| 4 Identify barriers and facilitators to change | 'Social/Professional Role, and identity' and 'Goals' | 'Skills,' 'Belief in Capabilities,' 'Reinforcement,' and 'Intentions' |
| 5 Identify intervention functions | 'Education,' 'Modelling,' and 'Enablement' | |
| 6 Identify policy categories | 'Communication/marketing' and 'Service provision' | |
| 7 Identify behavior change techniques | 'Action planning,' 'Goal setting (behavior),' 'Goal setting (outcome)' | Behavioral rehearsal/practice, 'Anticipation of future reward,' 'Behavioral contract,' 'Verbal persuasion' |
| 8 Identify delivery mode | Three integrated digital platforms, specifically Prolific, Qualtrics, and Twitter. | |

information anti-littering messages could contain, which were provided in the materials/procedures section of Diagnostic Survey 1.

Next, participants were randomly allocated to one of four groups, in a 1:1:1:1 fashion, including the Control, Goals, Social Identity-Positive, or Social Identity-Life Roles group:

**Control group.** Participants advanced in the survey without experiencing an intervention.

**Goals group.** Participants experienced three behavior change techniques. First, they were given a goal to tweet at least three anti-littering messages in the next seven days: the 'Goal-setting (behavior)' technique. Second, they described a positive outcome that might occur if they posted anti-littering messages: the 'Goal-setting (outcome)' technique. Third, they stated when, where, and in what circumstances they would post the anti-littering messages, along with three to seven ideas for future posts: the 'Action plan' technique.

**Social Identity-Positive group.** Participants experienced the 'Imaginary reward' technique. Specifically, they were encouraged to think about and describe the positive effects of posting anti-littering messages on Twitter in a free-text box.

**Social Identity-Life Roles group.** Participants experienced the 'Valued Self-Identity' technique. Specifically, they listed their three most important life roles (e.g., being a parent) and to rank those roles from the most to least important. They then read a passage about how these life roles influence behavior and described how posting anti-littering messages could help them better serve their most important life role in a free-text box.

Next, all participants wrote an anti-litter message with the hashtag #LetsUnlitterUK in a free-text box. Then they answered questions about their gender (Male, Female, Other, or Prefer not to say) and age. Finally, participants learned that they could post on Twitter by clicking a "Tweet" button, see the left side of Fig 1. Clicking this button opened a Twitter pop-up window. Those participants already logged into Twitter could click the "Tweet" button to post their message, see the right side of Fig 1. Those participants not already logged in saw a "Log in and Tweet" button and after logging in could Tweet their message. The final survey question asked about participants' intentions to post, "How many times, in the next seven days, do you intend to post an anti-littering message on Twitter?" (Zero, Once, Twice, Three, Four, Five, or more times).

A Google form was set up that imported all tweets using the provided hashtag. The imported tweets were matched to the message participants provided on Qualtrics using Excel's

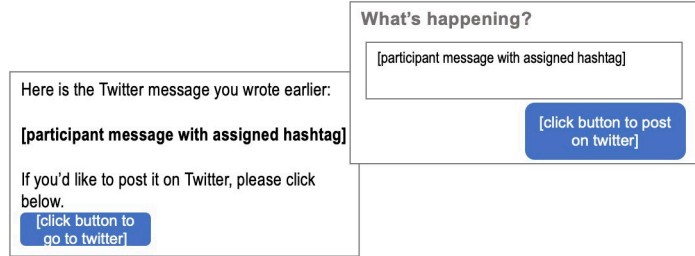

**Fig 1. Images showing what participants saw as they proceeded to tweet their anti-litter messages.**

matching function. A researcher and co-author (JK) manually scanned unmatched tweets to locate additional matches, e.g., resolving slight differences in spelling and punctuation. As a reminder, at this time we were unsure whether this matching method would work. Therefore, this outcome is used in an exploratory capacity to assess the fidelity of the retrieval process for the studies described in Section 2.

## Analyses

Descriptive statistics were compared to assess whether the groups were composed of participants with similar attitudes about litter and similar levels of activity on Twitter. Next Kruskal-Wallis tests were used to compare each group's intentions to post an anti-littering message. Then exploratory Chi-square tests were conducted to compare the percentage of participants who actually posted in each group. The significance of each comparison was assessed using a 0.05 alpha level, with Bonferroni's correction applied to post-hoc comparisons.

## Results: Evaluative Trial 1

While 1203 participants consented to take part in the survey, 133 did not complete the survey and a further 90 did not use Twitter and were removed from further analyses. Of the remaining 980 participants, 666 (68.0%) identified as female, 310 (31.6%) as male, 4 (0.4%) as other/preferred not to say. The mean age was 34.82 years (SD = 10.04). The average participant completed the survey in about five minutes (M = 4.96, SD = 3.39). Table 4 presents participants' demographics overall and for each group.

Regarding group allocation, 221 participants were allocated to the Goals group, 259 to the Social Identity-Positive group, 250 to the Social Identity-Life Roles group, and 250 to the Control group. Participants' attitudes towards litter were similar across groups with means ranging from 4.29 to 4.37. In addition, they posted on Twitter similar amounts with means ranging from 3.26 to 3.50 times a week.

**Table 4. Trial 1: Participant Demographics/Outcome: Evaluative Trial 1 about behavioral intentions.**

| Demographic/ Outcome | All | Goals | Social Identity–Positive | Social Identity–Life Roles | Control |
|---|---|---|---|---|---|
| Number (% of total) | 980 (100%) | 221 (22.6%) | 259 (26.4%) | 250 (25.5%) | 250 (25.5%) |
| Female (% of group) | 666 (68.0%) | 152 (68.8%) | 169 (65.3%) | 165 (66.0%) | 180 (72.0%) |
| Mean Age in Years (SD) | 34.82 (10.04) | 34.96 (9.43) | 34.14 (10.25) | 34.78 (10.07) | 35.44 (10.32) |
| Mean Attitude Towards Littler (SD) | 4.33 (0.73) | 4.32 (0.69) | 4.29 (0.75) | 4.37 (0.71) | 4.34 (0.75) |
| Mean Twitter Frequency (SD) | 3.37 (1.37) | 3.43 (1.32) | 3.26 (1.41) | 3.32 (1.35) | 3.50 (1.40) |
| Intentions (SD) | 3.21 (1.19) | 3.56 (1.12) | 3.07 (1.25) | 3.06 (1.17) | 3.19 (1.14) |
| Actual Tweets (% of group) | 188 (19.2%) | 50 (22.6%) | 42 (16.2%) | 48 (19.2%) | 48 (19.2%) |

Participants' intentions to post their anti-littering messages were highest in the Goals group ($M$ = 3.56, $SD$ = 1.12), followed by the Control ($M$ = 3.19, $SD$ = 1.14), Social Identity-Life Roles ($M$ = 3.07, $SD$ = 1.25), and Social Identity-Positive ($M$ = 3.06, $SD$ = 1.17) groups. A statistically significant difference was found between the groups' intentions to post their anti-littering message, $H(3)$ = 28.32, $p < .001$. The Goals group significantly differed from all other groups (all $p$'s $< .001$), and the remaining groups did not significantly differ from each other (all $p$'s = 1.00).

The Google doc retrieved 218 relevant posts from Twitter. The researcher was able to match 188 (86.24%) posts to the messages participants wrote on Qualtrics. Likely, not all of the posts could be matched because some participants changed their messages when they actually posted on Twitter. To improve matching in future surveys/trials, the researchers revised their instructions to ask participants to post the exact message they gave in Qualtrics on Twitter.

An exploratory analysis was then used to see if the percentage of participants who actually posted their anti-littering messages differed across groups. The percentage who posted was highest in the Goal group ($N$ = 50, 22.62%), followed by Control and Social Identity-Life Roles groups (both $N$'s = 48, 19.20%), and lastly the Social Identity-Positive group ($N$ = 42, 16.22%). A Chi-squared test revealed no differences between the percentage of participants who posted in each group, $X^2(3)$ = 3.16, $p$ = .37, $\varphi$ = 0.06.

## Brief discussion

Section 1 had two main objectives: to describe the creation of an intervention using the Behavior Change Wheel and to describe an evaluation of that intervention's effects on people's intentions to post anti-littering messages. Regarding the first objective, the goal intervention was completely informed by the Behavior Change Wheel. Regarding the second objective, the goal intervention positively influenced participants' intentions to post, while the interventions less or not at all informed by the Behavior Change Wheel did not.

The exploratory analyses suggest that no interventions influenced participants' actual posting. Data were not collected about whether or how much participants adhered to the intervention instructions. Future studies could examine the potential effectiveness of these interventions when participants adhere to the instructions in a laboratory setting. The current study's aims and analyses are more closely related to what is called "intention-to-treat" in clinical trials [31]. Practically, it is difficult to see how interventionists could ensure people sufficiently engage with an internet-delivered intervention, and, therefore, this limitation does not preclude advancing to the following studies also looking at interventions' practical effectiveness in real-world settings.

Another potential reason the interventions did not influence participants' posting is that we specified the target behavior as intentions to post. Previous research suggests that behavioral intentions are unreliable predictors of behavior [32,33]. But having established a method to measure actual posting behavior, Section 2 focuses on actual posting.

## Section 2: Increasing actual anti-littering tweets

Section 2 has two main objectives. The first is to develop an intervention using the Wheel methodology to increase actual anti-littering message posting. The second objective is to evaluate that intervention's effectiveness. As in Section 1, in Section 2, each step of the Behavior Change Wheel is described. The rightmost column of Table 3 summarizes our decisions at each step. The problem was defined in the same way (step 1), and the selected (step 2) and specified target behavior (step 3) was revised to focus on actual posts. Next, to diagnose the

reason(s) people are not already posting anti-littering messages (step 4), a quantitative method was used.

## Methods: Diagnostic Survey 2

### Participants

Participants completed the survey on the 5<sup>th</sup> or 6<sup>th</sup> of July 2018 and received 0.75 GBP. Those who participated in previous surveys were not invited to participate. The sample size for this diagnostic survey was increased to over 1000, which aligns with Bujang's et al.'s rule of thumb for observational studies and real-world data, where the sample size should be at least 500 plus 50 times x, where x is the number of predictor variables [34].

### Materials/Procedures

First, participants gave their informed consent and then indicated whether they used Twitter (Yes or No) and how much of a problem they thought litter was in the United Kingdom (1 = strongly agree to 7 = strongly disagree). Then participants were informed that they would be asked how much they agreed with statements about littering (1 = strongly to 7 = strongly agree) and were given the same examples of anti-littering messages as in Diagnostic Survey 1. Next, 28 statements expressing barriers and facilitators people experience to posting anti-littering messages appeared in a random order (see S1 Appendix). As in Diagnostic Survey 1, the statements were informed by the Theoretical Domains Framework [23,24]. Each domain was composed of two statements.

Next, participants stated how often they used Twitter (Never, Yearly, Monthly, Weekly, or Daily) and to write an anti-littering tweet with the hashtag #LetsUnlitterUK in a free-text box. Then they provided their gender (Male, Female, or Other/Prefer not to say) and age. Lastly, they were asked to tweet their messages using the same procedure as in Evaluative Trial 1.

Participants' actual tweets were retrieved using the same method described in Evaluative Trial 1.

### Analyses

Cronbach's alphas were calculated to assess the reliability of each domain's measures. As in Diagnostic Survey 1, domains with an alpha of less than .50 were split into multiple single-item predictors. The mean composite score for each domain was calculated using responses to its associated statements, or statement for single items. Then a logistical regression was performed with the Actual Tweets (Yes or No) as the outcome variable and the remaining domains as predictor variables. The significance of each domain was assessed using a 0.05 alpha level.

## Results: Diagnostic Survey 2

Of the 1012 participants, 623 (60.2%) identified as female, 383 (37.0%) identified as male, and 6 (0.6%) said other/preferred not to say. The mean age was 32.16 years ($SD$ = 10.88): one participant did not provide their age. The average participant completed the survey in about six minutes ($M$ = 6.13, $SD$ = 4.31). Of the 1012 participants, 100 (9.9%) actually posted their tweets.

As the following domain's Cronbach alphas were all less than the pre-set threshold of .50, their items were split into multiple single-item predictors: 'Behavioral Regulation' (.43), 'Social Influences' (.49), and 'Environmental contexts and resources' (.43). The remaining composite

domain scores' Cronbach alphas ranged from .52 to .92. Table 5 presents each domain's number of statements retained, Cronbach's alpha, and mean composite score.

Next, a regression analysis was performed to understand how actual posting behavior was influenced by the remaining domains. The overall model was significant, $X^2(17) = 94.57$, $p < .001$, Nagelkerke $R^2 = 0.19$. The following five domains significantly contributed to the model at the predetermined 0.05 alpha level: 'Skills' (*Odds ratio* = 1.51, *p* = 0.05, *95% confidence interval* [1.02, 2.23]), Social influences–Item 2 (*Odds ratio* = 0.80, *p* = .01, *95% confidence interval* [0.83, 1.33], 'Beliefs about capabilities' (*Odds ratio* = 1.42, *p* = .02, *95% confidence interval* [1.07, 1.89]), 'Intentions' (*Odds ratio* = 1.74, *p* < .001, *95% confidence interval* [1.37, 2.21]), and 'Reinforcement' (*Odds ratio* = 0.74, *p* = .01, *95% confidence interval* [0.59, 0.92]). Table 6 displays the results of the regression, along with each domain's contributions to the model.

## Brief discussion

Diagnostic Survey 2 identified five domains that significantly influence whether people actually post anti-littering messages: 'Skills,' 'Social influences–item 2,' 'Beliefs about capabilities,' 'Intentions,' and 'Reinforcement.' As previously noted, a single-component intervention focused on a single barrier may prove insufficient to change behavior, and so we aimed to develop a multi-component intervention focused on the four domains simultaneously in the future intervention. Using these four domains, we selected the same intervention functions (step 5) and policy categories (step 6) in Section 2 as in Section 1.

The behavior change techniques selected (step 7) were adjusted to align with the identified domains to create a multi-component intervention. Table 7 shows all the behavior change techniques considered. One technique from each identified domain was selected: the 'behavioral rehearsal/practice' technique for the 'Skills' domain; the 'verbal persuasion to boost self-efficacy' technique for 'Beliefs in capabilities'; the 'behavioral contract' technique for 'Intentions'; and 'anticipation of future reward' for 'Reinforcement'. Lastly, we selected the same mode of delivery (step 8) in Section 2 as in Section 1. Section 2 now describes the randomized

**Table 5. Cronbach's alpha and composite scores for the Theoretical Domains Framework: Diagnostic 2 survey about actual behavior.**

| Domain | Number of statements retained | Cronbach's Alpha | Mean Composite Score |
|---|---|---|---|
| Knowledge | 2 | 0.69 | 5.48 |
| Skills | 2 | 0.55 | 6.04 |
| Memory, attention, and decision processes | 2 | 0.52 | 4.26 |
| Behavioral regulation–item 1 | 1 | n/a | 3.02 |
| Behavioral regulation–item 2 | 1 | n/a | 4.71 |
| Social influences–item 1 | 1 | n/a | 4.17 |
| Social influences–item 2 | 1 | n/a | 2.92 |
| Environmental contexts and resources–item 1 | 1 | n/a | 6.67 |
| Environmental contexts and resources–item 2 | 1 | n/a | 6.32 |
| Social/professional role and identity | 2 | 0.65 | 3.47 |
| Beliefs about capabilities | 2 | 0.66 | 4.92 |
| Optimism | 2 | 0.81 | 4.34 |
| Intentions | 2 | 0.92 | 2.88 |
| Goals | 2 | 0.67 | 4.14 |
| Beliefs about consequences | 2 | 0.76 | 4.33 |
| Reinforcement | 2 | 0.80 | 3.21 |
| Emotions | 2 | 0.88 | 4.51 |

**Table 6. Logistical regression predicting Actual Behavior from domain composite scores: Diagnostic 2 survey about actual behavior.**

| | b | SE(b) | p-score | Odds Ratio | 95% CI | |
|---|---|---|---|---|---|---|
| (Constant) | -6.65 | 1.28 | < .001 | 0.00 | | |
| Knowledge | -0.10 | 0.13 | 0.43 | 0.90 | 0.70 | 1.17 |
| Skills | 0.41 | 0.20 | 0.04* | 1.51 | 1.02 | 2.23 |
| Memory, attention and . . . | -0.08 | 0.11 | 0.47 | 0.92 | 0.74 | 1.15 |
| Beh reg–item 1 | 0.10 | 0.09 | 0.27 | 1.11 | 0.92 | 1.33 |
| Beh reg–item 2 | -0.14 | 0.09 | 0.10 | 0.87 | 0.73 | 1.03 |
| Social influences–item 1 | 0.05 | 0.12 | 0.68 | 1.05 | 0.83 | 1.33 |
| Social influences–item 2 | -0.22 | 0.08 | <0.01** | 0.80 | 0.69 | 0.94 |
| Environmental . . .– item 1 | -0.19 | 0.11 | 0.08 | 0.82 | 0.66 | 1.02 |
| Environmental . . . –item 2 | 0.20 | 0.16 | 0.21 | 1.23 | 0.89 | 1.69 |
| Social/professional role. . . | 0.04 | 0.14 | 0.77 | 1.04 | 0.80 | 1.36 |
| Belief in capabilities | 0.35 | 0.15 | 0.02* | 1.42 | 1.07 | 1.89 |
| Optimism | -0.18 | 0.17 | 0.29 | 0.84 | 0.61 | 1.16 |
| Intentions | 0.55 | 0.12 | <0.01** | 1.74 | 1.37 | 2.21 |
| Goals | 0.04 | 0.13 | 0.73 | 1.05 | 0.81 | 1.35 |
| Belief in consequences | 0.12 | 0.18 | 0.51 | 1.13 | 0.79 | 1.60 |
| Reinforcement | -0.30 | 0.11 | <0.01** | 0.74 | 0.59 | 0.92 |
| Emotions | 0.17 | 0.16 | 0.28 | 1.19 | 0.87 | 1.62 |
| Nagelkerke $R^2$ | 0.19 | | | | | |
| $X^2$ | 94.57** | | | | | |

*significant at $p < 0.05$.

**significant at $p < .01$.

controlled trial conducted to evaluate the effectiveness of this multi-component intervention compared to an intervention based on a social norms approach and a no-intervention control group.

## Methods: Evaluative Trial 2

### Participants

Participants completed the survey on the 5th or 8th of August 2018 and received 0.50 GBP. The primary dependent measure was the proportion of participants who posted their messages on Twitter. To power this analysis, we would need at least 241 participants in each group to

**Table 7. Links between the identified Theoretical domains and the behavior change techniques.**

| Theoretical Domain | Behavior change techniques |
|---|---|
| • Skills | Behavioral rehearsal/practice[a]; Body changes; Graded tasks; Habit formation; Habit reversal |
| • Beliefs about capabilities | Focus on past success; Verbal persuasion to boost self-efficacy[a] |
| • Reinforcement | Anticipation of future rewards or removal of punishment[a]; Classical conditioning; Counter conditioning; Differential reinforcement; Discrimination training; Extinction; Incentive; Material reward; Negative reinforcement; Non-specific reward; Punishment; Response cost; Self-reward; Shaping; Social reward; Threat; Thinning |
| • Intentions | Behavioral contract[a]; Commitment |

[a] Indicates the techniques selected for Evaluative Trial 2.

detect a 10% increase (from 10% to 20%) with 80% power, and an alpha of 0.025 which is Bonferroni's correction applied for multiple comparisons. A sample size of 382 in each group would raise our power to 95%; 1421 participants were retained in our analyses with 293 to 598 participants in each group.

## Materials/Procedures

First, participants gave their informed consent and then indicated whether they used Twitter (Yes or No). Next, they were informed that littering was a significant issue in the United Kingdom and that we would ask them to write an anti-littering message that they could post on Twitter. Participants were given the same examples of anti-littering messages as in Diagnostic Survey 1. Next, participants were randomly allocated to one of three groups in a 2:2:1 fashion: the Control, Multi-component, and Social Norms groups. Equal allocation was set for the Control and Multi-component group, as the chief practical aim of the study was to develop and evaluate interventions informed by the Wheel methodology. The opportunity to evaluate an intervention not at all informed by the Wheel methodology was considered later, and we decided that the addition of the Social Norms would offer interesting, though more exploratory comparisons to inform future studies. As fewer participants are allocated to the Social Norms group its outcomes will be less precise and should be interpreted more cautiously [35]. What participants experienced in each group is described below.

**Control group.** Participants simply advanced in the survey.

**Multi-component group.** Participants experienced four behavior change techniques informed by the Behavior Change Wheel. Regarding the 'Skills' domain, participants were asked to consider an example of posting using a two-step tweet process as if they were practicing posting, see Fig 1, and indicated whether they understood the instructions (Yes or No): the 'Behavioral rehearsal/practice' technique. Regarding the 'Reinforcement' domain, participants were told that if they actually posted their anti-littering message, we would email them "Five Tips on How to Spend Money to Increase Your Happiness" and indicated if they would want this email (Yes or No): the 'Anticipation of future reward' technique. Regarding the 'Intentions' domain, participants typed their initials under a commitment statement that read, "I will tweet an anti-littering message at the end of this study to help raise awareness of the problem of litter": the 'Behavioral contract' technique. Regarding the 'Belief in capabilities' domain, participants saw a picture of a child pumping their fist and saying, "Hey, you can do it!" before composing their anti-littering message: the 'Verbal persuasion to boost self-efficacy' technique.

**Social Norms group.** Participants saw a short message about how many people tweeted in the previous studies, "In our previous studies, we asked Prolific Academic users just like you to post on Twitter their anti-littering messages, which included the hashtag #LetsUnlitterUK. In response, close to 200 people tweeted!" This group allows us to test whether a more straightforward social norms approach increases participants' posts, without additional insights from the Behavior Change Wheel.

Next, participants wrote an anti-littering message exactly how they would want it displayed on Twitter with an already typed-in hashtag. All hashtags include the same number of characters, a negating prefix, a capitalized first letter, the word litter, and capitalized final two letters. The hashtags for the Multi-component and Control groups were counterbalanced. The Multi-component group's hashtags were either #DelitterUK or #NolitterGB, and the Control group's messages were either #DelitterGB or #NolitterUK. The Social Norms group's hashtag was #UnlitterUK. Then participants were asked to post their message using the same method as described in Evaluative Trial 1, see Fig 1. Participants who did not want to tweet could skip to the next question.

Then participants provided their gender (Male, Female, Other, or Prefer not to say), age, and how often they used Twitter (Never, Yearly, Monthly, Weekly, or Daily). Lastly, they indicated how much they agreed that litter was a problem in the United Kingdom (1 = strongly disagree to 5 = strongly agree).

Participants' actual tweets were retrieved using the method described in Evaluative Trial 1. The sample size was informed by rules of thumb for market research, and product testing to have more than 200 participants in each group [36].

## Analyses

The descriptive statistics were compared across groups. A Chi-squared test of independence was used to compare the percentage of participants in each group who actually posted their message on Twitter. An alpha level of 0.05 was used to assess whether the differences were significant, and Bonferroni's correction was applied to assess post-hoc comparisons. From diagnostic survey 2's results, we knew that the percentage of participants who actually tweet could be relatively low (10%), and the present trial is not adequately powered for to test for interactions or to assess individual differences.

Additional analyses narrow in on the Multi-component group. The percentages of participants who endorsed each intervention technique ('Skills,' 'Reinforcement,' and 'Intentions') is provided. Then an exploratory logistic regression is performed to understand those techniques influence on participants' Actual Behavior (Yes, No), via a logistic regression.

## Results: Evaluative Trial 2

While 1558 participants completed the survey, 134 said they did not use Twitter and 3 did not complete all items. Of the remaining 1421 participants, 979 (68.89%) identified as female, 438 (30.76%) as male, and 4 (0.03%) as other/preferred not to say. The mean age was 37.83 years ($SD$ = 11.58). The average participant completed the survey in about four minutes ($M$ = 4.38, $SD$ = 11.21). Participants' attitudes toward litter were relatively stable across groups with means ranging from 6.13 to 6.17. In addition, participants' use of Twitter was relatively stable across groups with means ranging from 2.89 to 3.03. Table 8 shows demographics and outcomes across groups.

Regarding group allocation, 530 participants were allocated to the Multi-component group, 293 to Social Norms, and 598 to Control. The percentage who tweeted their message was highest in the Multi-component group (22.64%), followed by the Social Norms (11.30%) and Control (7.86%) groups. An overall Chi-squared test was significant, $X^2(2)$ = 53.18, $p < .001$, $\varphi$ = 0.19. Post-hoc comparisons revealed a significant difference between the Multi-component and Control groups, $X^2(1)$ = 48.68, $p < .001$, $\varphi$ = 0.21, and between the Multi-component and

**Table 8. Experiment 2-Participant Demographics/Outcome: Evaluative Trial 1 about actual behavior.**

| Demographic/ Outcome | All | Multi-component | Social Norms | Control |
|---|---|---|---|---|
| Number (% of total) | 1421 | 530 (37.3%) | 293 (20.6%) | 598 (42.1%) |
| Female (% of group) | 979 (68.90%) | 373 (70.38%) | 210 (71.76%) | 396 (66.22%) |
| Mean Age in Years (SD) | 37.83 (11.58) | 38.09 (11.43) | 37.76 (11.57) | 37.63 (11.73) |
| Mean Attitude Towards Littler (SD) | 6.15 (0.89) | 6.15 (0.89) | 6.17 (0.86) | 6.13 (0.89) |
| Mean Twitter Frequency (SD) | 2.93 (1.39) | 2.89 (1.39) | 3.03 (1.39) | 2.91 (1.38) |
| Actual Post (%) | 200 (14.07%) | 120 (22.64%)* | 33 (11.30%) | 47 (7.86%) |

*significant from both other groups at $p < .001$.

Social Norms groups, $X^2(1) = 16.21$, $p < .001$, $\varphi = 0.14$. The difference between the Social Norms and Control groups was not significant, $X^2(1) = 2.79$, $p = .10$, $\varphi = 0.06$.

Of the 530 participants in the Multi-component group, 259 (48.87%) endorsed all three intervention components. Five (0.94%) indicated that they did not understand the directions, 211 (39.81%) did not want a happiness email, and 158 (29.81%) did not initial the behavioral contract. The overall model was significant, $X^2(3) = 68.18$, $p < .001$, Nagelkerke $R^2 = 0.18$. Hosmer-Lemeshow test of goodness of fit was not significant, $X^2(3) = 6.34$. $p = .10$. All components were effective at the 0.05 alpha level. The 'Skills' component was a negative contributor (*Odds ratio* = 0.08, $p = .03$, *95%* confidence interval [0.01, 0.78]): though, the reader should recall that only 5 participants did not endorse this items. The 'Reinforcement' component (*Odds ratio* = 1.82, $p = .02$, *95%* confidence interval [1.11, 2.99]) and the 'Intentions' component (*Odds ratio* = 9.98, $p < .001$, *95%* confidence interval [4.16, 23.91]) were positive contributors.

## Discussion

The current article showed how the Behavior Change Wheel can be used to develop effective interventions that promote intended and actual pro-environmental behaviors. Section 1 described how a goal-based intervention informed by the Behavior Change Wheel increased people's intentions but not actual posting behaviors. Section 2 described a successful intervention to increase people's sharing of anti-littering messages. A significantly greater percentage of participants who experienced the multi-component intervention completely informed by the Wheel (23.5%) posted their message than participants who experienced a more straightforward social norms approach (11.5%), or no intervention (7.9%).

The present study had practical aims: to develop and evaluate interventions that stand to increase pro-environmental posts using the Behavior Change Wheel methodology. Future studies may explore what individual differences influence the intervention's effectiveness and what components contribute most strongly to its effectiveness using newly collected data or reanalyzing our available data (**https://figshare.com/articles/dataset/_LetsUnlitterUK_A_ demonstration_and_evaluation_of_the_Behavior_Change_Wheel_Methodology/ 16895335**). Our participants were slightly younger (30–38 years old on average) than the United Kingdom's population's average of 40 years, and more females than male participants took part [37]; this may limit the generalizability of our findings.

Another possible limitation is its focus on digital-world behavior. Developing and evaluating an intervention about actual littering behavior may have been more difficult and more convincing. Unfortunately, the effectiveness of anti-littering interventions is difficult to measure and most evaluations rely on self-reports of past behavior, self-reports of intentions, or improved knowledge [38]. We focused on social media behavior, in part, because we wanted to evaluate the effectiveness of the intervention using a more rigorous randomized controlled trial where participants could be individually allocated to different conditions and where actual behavior could be measured. Further, digital-world behaviors are real-world behaviors. Statista 2019 estimates that in 2019 Twitter had 330 million monthly active users worldwide [39]. Further, it is accepted that social media can be used to change offline behavior [40,41]. For example, an analysis of the 2016 presidential election in the United States of America found a strong influence of Twitter and Facebook on voting behavior [42].

Evaluation Trial 2's interventions could be construed as 'more complex' and 'less complex'. A future trial may control for intervention complexity by equating the number of behavior change techniques used across interventions completely informed by the Behavior Change Wheel and only partially informed by it. Still, it should be noted that using the Behavior

Change Wheel methodology led us to create a more complex intervention that addressed multiple types of barriers, whereas using a more straightforward social norms approach led to less complex intervention. Therefore, if it is discovered that more complex interventions are typically more effective, the Behavior Change Wheel methodology may still prove a useful guide to select multiple techniques. Additional limitations for Evaluation Trial 2 involve the hashtags. Hashtags including "GB" may not have felt inclusive for any participants residing in Northern Ireland, which is not part of Great Britain. As the population of Northern Ireland makes up approximately 3% of the United Kingdom's population any effects were likely minor. As the hashtag for the Social Comparison group was not counterbalanced with the other groups and fewer participants were allocated to this group, our findings for the Social Norms group should be interpreted with greater caution.

The present article is the first to jointly describe the development and evaluation of interventions using the Behavior Change Wheel. Michie et al.'s *The Behavior Change Wheel*: *A Guide to Designing Interventions* purposefully does not describe how to evaluate interventions but directs readers to the Medical Research Council's complex intervention development and evaluation framework [1,43]. Possibly, as a consequence of this choice, the Behavior Change Wheel is well-described in many articles where formative research is conducted to inform the content of future interventions but not in those which evaluate interventions. Examples include interventions to increase guideline adherence [44,45], to increase safe-sex behaviors [46,47], to increase physical activity [48,49], and to decrease tobacco use [50,51]. Any full evaluations of these interventions' effectiveness (if they exist) can be difficult to find; consequently, readers who do not already have faith in the Behavior Change Wheel's ability to create effective interventions may remain skeptical. To mitigate this problem, trial pre-registrations can help readers link related formative and evaluative publications [for example, see reference 48], but the practice of pre-registering trials is still relatively uncommon outside health-focused randomized controlled trials. The present article should help mitigate skepticism by presenting the simultaneous development and evaluation of behavior change interventions.

Another strength of the present research is its application of the Behavior Change Wheel outside of health-related behavior. While conceptually the Behavior Change Wheel applies to any behavior, in practice its use is largely restricted to health-related behaviors. Note that all the studies mentioned in the previous paragraph are about health-related behaviors. In 2016, Gainforth, et al. demonstrated how the Behavior Change Wheel could be used to develop interventions to promote recycling behaviors through a series of structured interviews [52]. As in the current Diagnostic Survey 2, they also found a strong influence of the 'Intentions' domain but also found an influence of the 'Environmental context and resources,' 'Beliefs about consequences,' and 'Knowledge' domains. The fact that they found different barriers/facilitators need not be surprising, because how a behavior is defined (as intentions to post, actual posting behavior, or actual littering behavior) may influence which domains are relevant.

When developing and evaluating future interventions, some pro-environmental behaviors will likely be easier to measure than others. For example, Lacasse assessed whether participants wrote a pro-environmental message to their local government [53], and Carrico et al. measured how much participants donated to a pro-environmental charity [54]. Where objective measures are not available, self-report may suffice. For example, Wallis and Klöckner measured self-reported energy consumption [55], Supakata measured knowledge about recycling [56], and Sintov et al. measured whether participants reported composting leftover food [57]. Where possible, creative techniques should be encouraged to objectively measure actual behaviors. For example, objective measures of household energy use often are available via a household meter, the amount of waste disposed of in recycling bins can be weighed, and whether or not participants start composting could be observed via household visits.

In conclusion, the present article demonstrates how to apply the Behavior Change Wheel to encourage pro-environmental behaviors. The findings of the Evaluative Trials suggest that interventions developed using this methodology are more likely to be effective than interventions not so informed. The discussion encourages future use of the Behavior Change Wheel methodology, particularly beyond of health-related behaviors.

## Supporting information

**S1 Appendix.**
(DOCX)

## Author Contributions

**Conceptualization:** Julia Kolodko, Kelly Ann Schmidtke, Daniel Read, Ivo Vlaev.

**Data curation:** Julia Kolodko.

**Formal analysis:** Julia Kolodko, Kelly Ann Schmidtke, Daniel Read.

**Investigation:** Julia Kolodko.

**Methodology:** Julia Kolodko, Daniel Read.

**Project administration:** Julia Kolodko.

**Resources:** Julia Kolodko.

**Software:** Julia Kolodko.

**Supervision:** Daniel Read, Ivo Vlaev.

**Writing – original draft:** Julia Kolodko.

**Writing – review & editing:** Kelly Ann Schmidtke, Daniel Read, Ivo Vlaev.

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
