## [Decision Letter · Decision Letter 0]

1 Sep 2021

PONE-D-21-24044

#LetsUnlitterUK: A demonstration and evaluation of the Behavior Change Wheel Methodology

PLOS ONE

Dear Dr. Schmidtke,

Thank you for submitting your manuscript to PLOS ONE. After careful consideration, we feel that it has merit but does not fully meet PLOS ONE’s publication criteria as it currently stands. Therefore, we invite you to submit a revised version of the manuscript that addresses the points raised during the review process.

We look forward to receiving your revised manuscript.

Kind regards,

Prof. Anat Gesser-Edelsburg, Ph.D.

Academic Editor

PLOS ONE

Journal Requirements:

2. PLOS ONE has specific requirements for studies using personal data from third-party sources, including social media, blogs, other internet sources, and phone companies (https://journals.plos.org/plosone/s/submission-guidelines#loc-personal-data-from-third-party-sources). These requirements include confirming data are collected and used in accordance with the company or website’s Terms and Conditions, obtaining appropriate ethics or data protection body review, and ensuring appropriate consent from individuals whose data are used in research. In this case, please ensure that your Ethics statement is in compliance with guidelines, and that you have complied with the company's (i.e., Facebook's) Terms and Conditions, with appropriate permissions.

[This project was supported by the National Institute for Health Research (NIHR) Applied Research Centre (ARC) West Midlands. The views expressed are those of the author(s) and not necessarily those of the NIHR, ARC, or the Department of Health and Social Care. The funders had no role in the design of the study and collection, analysis, and interpretation of data and in writing the manuscript.]

 [This project was supported by the National Institute for Health Research (NIHR) Applied Research Centre (ARC) West Midlands. The views expressed are those of the author(s) and not necessarily those of the NIHR, ARC, or the Department of Health and Social Care. The funders had no role in the design of the study and collection, analysis, and interpretation of data and in writing the manuscript.]

5. We note that Figure 1 in your submission contain copyrighted images. All PLOS content is published under the Creative Commons Attribution License (CC BY 4.0), which means that the manuscript, images, and Supporting Information files will be freely available online, and any third party is permitted to access, download, copy, distribute, and use these materials in any way, even commercially, with proper attribution. For more information, see our copyright guidelines: http://journals.plos.org/plosone/s/licenses-and-copyright.

a) You may seek permission from the original copyright holder of Figure 1 to publish the content specifically under the CC BY 4.0 license. 

6. Please include a copy of Table 8 which you refer to in your text on page 20.

7. We note you have included a table to which you do not refer in the text of your manuscript. Please ensure that you refer to Table 9 in your text; if accepted, production will need this reference to link the reader to the Table.

Reviewers' comments:

Reviewer's Responses to Questions

**Comments to the Author**

1. Is the manuscript technically sound, and do the data support the conclusions?

Reviewer #1: Yes

Reviewer #2: Partly

2. Has the statistical analysis been performed appropriately and rigorously? 

Reviewer #1: Yes

Reviewer #2: I Don't Know

3. Have the authors made all data underlying the findings in their manuscript fully available?

Reviewer #1: Yes

Reviewer #2: No

4. Is the manuscript presented in an intelligible fashion and written in standard English?

Reviewer #1: Yes

Reviewer #2: Yes

5. Review Comments to the Author

Reviewer #1: I really liked the paper, particularly because not only it introduces a new approach/method, it also tests it, it shows its success, and then it shows what parts of it work and what predicts its success. So, it does not look at a behavioral interevention as a black box, but as a process that can be enhanced. So, the process evaluation and the predictors are very informative and add to the value of the paper.  

My concern # 1 is: that the results are not reported based on race, ethnicity, education level, and sex. However, we see what predicts the outcome.

My concern # 2 is: that it is easier to change a behavior such as online posting, and it would be more difficult to change chronic disease management, adherence, or exercise. So, the readers should not assume that the same model would generate the same results for other behaviors that may be more challenging to modify.

Reviewer #2: The present study applies the Behavior Change Wheel as a framework to guide development of anti-littering interventions. Across 2 observational and 2 experimental studies, the authors show that various behavior change domains (e.g., skills, social identity) are associated with either intention to post anti-littering social or actual posting. Interventions designed to target these domains, in turn, increase intentions and actual posting.

This was a clever, albeit complex, series of studies which could have substantial impact. However, I have a number of major and minor concerns before this manuscript can be evaluated appropriately.

Major concerns

I agree with the authors’ speculation that the goal-setting, social identify-positive, and social identity-life roles interventions in Evaluative Study 1 may have failed to increase actual posting because the observational study that identified these interventions were focused on intentions. However, another possibility is that engagement with these interventions was not sufficiently high to promote actual behavior change. In each of these interventions, participants were asked to think about and describe various goals or outcomes associated with posting these messages. What was the degree of adherence to these instructions? Was quality of engagement assessed? Did quality predict posting outcomes? Without such secondary analyses, it’s hard to evaluate where these interventions failed in influencing actual posting behavior.

The hashtag participants were asked to use while posting appears to alternate between #UnlitterUK, #DelitterUK, and #NolitterGB. This is primarily concerning in Evaluative Study 2, in which different intervetion groups were instructed to use different hashtags. This is presumably to aid in identification of group-specific tweets, but could have had an influence on response rates (e.g., due to possible differences in appeal, ease of use, or geographical footprints (UK vs. GB)). A rationale should be provided for this choice and it should be discussed as a possible limitation.

In Evaluative Study 2, the authors used logistic regression to examine the possible influence of each component of a multi-component intervention on posting. Were interaction terms considered in this analysis? If so, results should be reported. If not, rationale should be provided. Interactions between different components may be important, as they may help identify individual components that are most or least effective in combination.

More rationale should be provided for exploring a multi-component intervention, rather than individual interventions.

No rationale is provided (to my reading) for unequal allocation to groups in Evaluative Study 2.

Minor concerns

In several places, interventions provided participants with anti-littering messages prior to having the opportunity to post or rate their intentions to post. These example posts should be provided as supplementary material or in text.

Although I recognize the value of bundling Studies 1 and 2 together in a single manuscript, doing so also increases complexity and limits the space required to provide readers with sufficient explanation of study procedures and analyses. I wonder (merely a suggestion) if these data sets would be more appropriately communicated as two separate manuscripts: one comprising Diagnostic Study 1 and Evaluative Study 1, and the other comprising Diagnostic Study 2 and Evaluative Study 2.

6. PLOS authors have the option to publish the peer review history of their article (what does this mean?). If published, this will include your full peer review and any attached files.

Reviewer #1: No

Reviewer #2: No

---

## [Author Response · Author response to Decision Letter 0]

5 Oct 2021

A word version of the revision letter is also available, which may be easier for the editors and reviewers to read. 

In this revision letter we describe how we responded to the editor’s and reviewers’ comments. Where possible we state where changes were made in the manuscript (with page and line numbers). All authors have approved these changes. 

Editor comments: 

Editor 1 comment 1. Please ensure that your manuscript meets PLOS ONE's style requirements, including those for file naming. The PLOS ONE style templates can be found at 

>>>Author Response to Editor 1 comment 1. Thank you for drawing our attention to the formatting samples. We have modified the text to include the new header structure throughout in Microsoft Word. 

Editor 1 comment 2. PLOS ONE has specific requirements for studies using personal data from third-party sources, including social media, blogs, other internet sources, and phone companies (https://journals.plos.org/plosone/s/submission-guidelines#loc-personal-data-from-third-party-sources). These requirements include confirming data are collected and used in accordance with the company or website’s Terms and Conditions, obtaining appropriate ethics or data protection body review, and ensuring appropriate consent from individuals whose data are used in research. In this case, please ensure that your Ethics statement is in compliance with guidelines, and that you have complied with the company's (i.e., Facebook's) Terms and Conditions, with appropriate permissions.

>>>Author Response to Editor 1 comment 2. We confirm that our ethics statement is in compliance with the guidelines, and we have complied with terms and conditions with appropriate permissions. 

Editor 1 comment 3. Thank you for stating the following in the Acknowledgments Section of your manuscript: 

[This project was supported by the National Institute for Health Research (NIHR) Applied Research Centre (ARC) West Midlands. The views expressed are those of the author(s) and not necessarily those of the NIHR, ARC, or the Department of Health and Social Care. The funders had no role in the design of the study and collection, analysis, and interpretation of data and in writing the manuscript.]

 [This project was supported by the National Institute for Health Research (NIHR) Applied Research Centre (ARC) West Midlands. The views expressed are those of the author(s) and not necessarily those of the NIHR, ARC, or the Department of Health and Social Care. The funders had no role in the design of the study and collection, analysis, and interpretation of data and in writing the manuscript.]

>>>Author Response to Editor 1 comment 3. The funding statement has been removed from our manuscript. The funding statement (now including a grant ID number) is provided within our cover letter, and we thank the editorial team for changing the online submission form on our behalf. 

Editor 1 comment 4. We note that you have stated that you will provide repository information for your data at acceptance. Should your manuscript be accepted for publication, we will hold it until you provide the relevant accession numbers or DOIs necessary to access your data. If you wish to make changes to your Data Availability statement, please describe these changes in your cover letter and we will update your Data Availability statement to reflect the information you provide.

>>>Author Response to Editor 1 comment 4. If accepted, we are happy for you to hold our paper until a DOI is provided for data access. Online will be both the data files and the SPSS outputs that informed the present manuscript. 

Editor 1 comment 5. We note that Figure 1 in your submission contain copyrighted images. All PLOS content is published under the Creative Commons Attribution License (CC BY 4.0), which means that the manuscript, images, and Supporting Information files will be freely available online, and any third party is permitted to access, download, copy, distribute, and use these materials in any way, even commercially, with proper attribution. For more information, see our copyright guidelines: http://journals.plos.org/plosone/s/licenses-and-copyright.

a) You may seek permission from the original copyright holder of Figure 1 to publish the content specifically under the CC BY 4.0 license. 

>>>Author Response to Editor 1 comment 5. Figure 1 has been remade and no longer contains any copyrighted images. A small version of the remade figure appears in the word version of this revision letter and in the revised manuscript. 

Editor 1 comment 6. Please include a copy of Table 8 which you refer to in your text on page 20.

And 

Editor 1 comment 7. We note you have included a table to which you do not refer in the text of your manuscript. Please ensure that you refer to Table 9 in your text; if accepted, production will need this reference to link the reader to the Table.

>>>Author Response to Editor 1 comment 6 and 7. The title of “Figure 9” has been changed to “Figure 8”. 

Reviewer 1 comments

Reviewer #1 comment. I really liked the paper, particularly because not only it introduces a new approach/method, it also tests it, it shows its success, and then it shows what parts of it work and what predicts its success. So, it does not look at a behavioral intervention as a black box, but as a process that can be enhanced. So, the process evaluation and the predictors are very informative and add to the value of the paper. 

>>>Author Response to reviewer 1 comment. Thank you. We are happy that this overall story come across in our manuscript. 

Reviewer #1 concern 1. My concern # 1 is: that the results are not reported based on race, ethnicity, education level, and sex. However, we see what predicts the outcome.

>>>Author Response to reviewer 1 concern 1. The goal of the current study was not to understand individual differences in responses to our interventions, and the study was not designed to adequately inform such analyses, e.g., about racial differences. Information about gender was collected but not analyzed. As pointed out by Reviewer 2, the paper is already quite complex, and we do not wish to increase its complexity with the addition of new analyses outside the paper’s main aims. The data will be made publicly available for other researcher who wish to undertake such analyses. The aims of the current study are now more clearly stated in the Introduction, staring on line 67:

The aims of the current study are practical: to develop and evaluate interventions that stand to increase pro-environmental posts online in the United Kingdom using the behavior change wheel methodology. We do not assess individual differences or particular mechanisms of action, e.g., to evaluate the effectiveness of components of the interventions.

In the discussion we also now state, starting at line 580:

Future studies may explore what individual differences influence the intervention’s effectiveness and what components contribute most strongly to its effectiveness.

Reviewer #1 concern 2. My concern # 2 is: that it is easier to change a behavior such as online posting, and it would be more difficult to change chronic disease management, adherence, or exercise. So, the readers should not assume that the same model would generate the same results for other behaviors that may be more challenging to modify.

>>>Author Response to reviewer 1 concern 2. We agree that it is likely easier to change online posting behavior than real-world littering. This is highlighted in the discussion, starting at line 587:

Another possible limitation is its focus on digital-world behavior. Developing and evaluating an intervention about actual littering behavior may have been more difficult and more convincing.

We also apologies that the manuscript did not previously emphasize how widely used and promoted the Behavior Change Wheel’s methodology for health-related research, at least in the United Kingdom. The Wheel was published 10 years ago (in 2011), and the manual was published in 2014. The development of the methodology was developed by a Professor of Health Psychology (Prof Susan Michie) who is the Director of the Centre for Behavior Change, University College London. The Wheel is frequently cited in studies aiming to development complex interventions to improve people’s health. The first sentence of the manuscript has been adjusted to make this clearer, starting at line 51:

While the Wheel methodology is already widely used to develop health-related interventions (National Institute for Health and Care Excellence, 2014; Michie and West, 2021), we are the first to jointly describe the development of interventions using the Behavior Change Wheel and the evaluation of those interventions’ effectiveness on intended and then actual behavior.

Reviewer 2 comments

Reviewer #2: comment 1. The present study applies the Behavior Change Wheel as a framework to guide development of anti-littering interventions. Across 2 observational and 2 experimental studies, the authors show that various behavior change domains (e.g., skills, social identity) are associated with either intention to post anti-littering social or actual posting. Interventions designed to target these domains, in turn, increase intentions and actual posting. This was a clever, albeit complex, series of studies which could have substantial impact. However, I have a number of major and minor concerns before this manuscript can be evaluated appropriately.

>>>Author Response to reviewer 2 comment 1. We agree that our paper could have a substantial impact on the way future interventions are designed. Thank you for your constructive comments that have surely improved the quality of the manuscript to achieve this impact. 

Reviewer #2 major concern 1. I agree with the authors’ speculation that the goal-setting, social identify-positive, and social identity-life roles interventions in Evaluative Study 1 may have failed to increase actual posting because the observational study that identified these interventions were focused on intentions. However, another possibility is that engagement with these interventions was not sufficiently high to promote actual behavior change. In each of these interventions, participants were asked to think about and describe various goals or outcomes associated with posting these messages. What was the degree of adherence to these instructions? Was quality of engagement assessed? Did quality predict posting outcomes? Without such secondary analyses, it’s hard to evaluate where these interventions failed in influencing actual posting behavior.

>>>Author Response to reviewer 2 major concern 1. This concern is now addressed in the brief discussion after the Evaluative Study 1, starting at line 357: 

Data were not collected about whether or how much participants adhered to the intervention instructions. Future studies could examine the potential effectiveness of these interventions when participants adhere to the instructions in a laboratory setting. The current study’s aims and analyses are more closely related to what is called “intention-to-treat” in clinical trials (Ranganathan et al., 2016). Practically, it is difficult to see how interventionists could ensure people sufficiently engage with an internet-delivered intervention, and, therefore, this limitation does not preclude advancing to the following studies also looking at interventions’ practical effectiveness in real-world settings.

Ranganathan, P., Pramesh, C. S., & Aggarwal, R. (2016). Common pitfalls in statistical analysis: Intention-to-treat versus per-protocol analysis. Perspectives in clinical research, 7(3), 144–146. https://doi.org/10.4103/2229-3485.184823

Reviewer #2 major concern 2. The hashtag participants were asked to use while posting appears to alternate between #UnlitterUK, #DelitterUK, and #NolitterGB. This is primarily concerning in Evaluative Study 2, in which different intervetion groups were instructed to use different hashtags. This is presumably to aid in identification of group-specific tweets, but could have had an influence on response rates (e.g., due to possible differences in appeal, ease of use, or geographical footprints (UK vs. GB)). A rationale should be provided for this choice and it should be discussed as a possible limitation.

>>>Author Response to reviewer 2 major concern 2. You are correct that different hashtags were used to identify group-specific tweets. To help ensure that participants used the assigned hashtags, the assigned hashtags were automatically populated in the in the post message, see the remade Figure 1. 

The hashtags linked to each group are more complicated than the comment suggests, and the text in the manuscript has been slightly revised to emphasize this point, starting at line 507: 

Next, participants wrote an anti-littering message exactly how they would want it displayed on Twitter with an already typed-in hashtag. All hashtags include the same number of characters, a negating prefix, a capitalized first letter, the word litter, and capitalized final two letters. The hashtags for the Multi-component and Control groups were counterbalanced. The Multi-component group’s hashtags were either #DelitterUK or #NolitterGB, and the Control group’s messages were either #DelitterGB or #NolitterUK. The Social Norms group’s hashtag was #UnlitterUK.

**a table describing these hashtags is provided in the word version of the response to reviewers**

The GB ending may have negatively influenced participants living in Northern Ireland – which is part of the United Kingdom, but not part of Great Britain. But these effects would have been slight. Northern Ireland makes up approximately 3% of the United Kingdom’s population represented in the National statistics and only 2% and these participants living in the United Kingdom in Proflic’s survey panel. This is now pointed out as a limitation in the discussion section. Additionally, the Social Norms group’s hashtag was not counterbalanced in the same manner as the Multi-component and Control groups. This is now pointed out as a limitation in the discussion section starting at line 609:

Additional limitations for Evaluation Trial 2 involve the hashtags. Hashtags including “GB” may not have felt inclusive for any participants residing in Northern Ireland, which is not part of Great Britain. As the population of Northern Ireland makes up approximately 3% of the United Kingdom’s population any effects were likely minor. As the hashtag for the Social Comparison group was not counterbalanced with the other groups and fewer participants were allocated to this group, our findings for the Social Norms group should be interpreted with greater caution.

Reviewer #2 major concern 3. In Evaluative Study 2, the authors used logistic regression to examine the possible influence of each component of a multi-component intervention on posting. Were interaction terms considered in this analysis? If so, results should be reported. If not, rationale should be provided. Interactions between different components may be important, as they may help identify individual components that are most or least effective in combination.

>>>Author Response to reviewer 2 major concern 3. No interaction terms were included in the analysis, as the main aim of the study was not to understand mechanisms, but rather more practically focused. As stated in your later comment (minor concern 2), the paper is already on the edge of being too complex and we do not wish to increase the complexity of the present analyses further. The data will be made publicly available for other researcher who have this particular interest, and we acknowledge this limitation in the discussion starting at line 580:

Future studies may explore what individual differences influence the intervention’s effectiveness and what components contribute most strongly to its effectiveness using newly collected data or reanalyzing our available data (FIG SHARE LINK ANON FOR PEER REVIEW).

Reviewer #2 major concern 4. More rationale should be provided for exploring a multi-component intervention, rather than individual interventions.

>>>Author Response to reviewer 2 major concern 4. Thank you for helping us explain this choice to evaluate a multi-component intervention a bit clearer. This explanation now starts in the introduction, where we more clearly state the study aims, starting on line 67: 

The aims of the current study are practical: to develop and evaluate interventions that stand to increase pro-environmental posts online in the United Kingdom using the behavior change wheel methodology. We do not assess individual differences or particular mechanisms of action, e.g., to evaluate the effectiveness of components of the interventions.

Then, the practical reason we developed multi-component interventions is further explained in the Brief Discussion of Diagnostic Survey 1, starting at line 185:

Moving forward in the intervention development process, we choose to only focus on the two most significant domains: ‘Goals’ and ‘Social/professional role and identity.’ More domains could have been selected, and in some cases would be required to achieve behavior change. For example, increasing national vaccination rates may require offering the vaccination in different potentially appealing locations (the “Environmental context and resources” domain), increasing awareness (the “knowledge” domain) that those opportunities exist, and overcoming negative feelings towards vaccinations (the “emotion” domain; Williams et al., 2020). The aims of the current study are narrower, aiming to influence only people who already have a Twitter account, and testing these two domains in combination presents a unique opportunity to ultimately evaluate intervention completely and partially informed by the Wheel methodology, as will be seen in step 7. But first, we address steps 5 and 6. 

Williams, L., Gallant, A. J., Rasmussen, S., Brown, Nicholls, L. A., Cogan, N., Deakin, K., Young, D., & Flowers, P. (2020). Towards intervention development to increase the uptake of COVID-19 vaccination among those at high risk: Outlining evidence-based and theoretically informed future intervention content. British Journal of Health Psychology, 25(4):1039–1054. https://doi.org/10.1111/bjhp.12468

Reviewer #2 major concern 5. No rationale is provided (to my reading) for unequal allocation to groups in Evaluative Study 2.

>>>Author Response to reviewer 2 major concern 5. This point is now addressed in Evaluative Trial 2’s methods, starting at line 477: 

Equal allocation was set for the Control and Multi-component group, as the chief practical aim of the study was to develop and evaluate interventions informed by the Wheel methodology. The opportunity to evaluate an intervention not at all informed by the Wheel methodology was considered later, and we decided that the addition of the Social Norms would offer interesting, though more exploratory comparisons to inform future studies. As fewer participants are allocated to the Social Norms group its outcomes will be less precise and should be interpreted more cautiously (Hey & Kimmelman, 2014). 

Hey, S. P. & Kimmelman, J. (2014). The questionable use of unequal allocation in confirmatory trials. Neurology, 82(1), 77–79. https://doi.org/10.1212/01.wnl.0000438226.10353.1c

Reviewer #2 minor concern 1. In several places, interventions provided participants with anti-littering messages prior to having the opportunity to post or rate their intentions to post. These example posts should be provided as supplementary material or in text.

>>>Author Response to reviewer 2 minor concern 1. Thank you for helping us increase the clarity of the text. Participants did not receive precise examples of anti-littering message, but rather examples for what the content of an anti-littering message could involve. We have revised the text in Diagnostic Survey 1 to be clearer about what the word “example” refers to, starting at line 131: 

Participants were provided with examples for what types of information anti-littering messages could contain: encouragement for others to not litter or to clean up litter; warnings about the negative consequences of litter; pictures of litter; or pictures of litterers/fly-tippers.

Reviewer #2 minor concern 2. Although I recognize the value of bundling Studies 1 and 2 together in a single manuscript, doing so also increases complexity and limits the space required to provide readers with sufficient explanation of study procedures and analyses. I wonder (merely a suggestion) if these data sets would be more appropriately communicated as two separate manuscripts: one comprising Diagnostic Study 1 and Evaluative Study 1, and the other comprising Diagnostic Study 2 and Evaluative Study 2.

>>>Author Response to reviewer 2 major concern 5. The research team discussed the option of splitting up the present manuscript into multiple shorter manuscripts. While we agree that this could make the story for each manuscript simpler, salami slicing the current manuscript could also decreases a more interesting and holistic story that builds across the current manuscript. To help readers split the paper into more discrete parts, the headers “section 1” and “section 2” are included. To help readers follow the story line, we refrain from complex analyses for potentially interesting but more tangential areas of interest, e.g., individual differences.

---

## [Decision Letter · Decision Letter 1]

26 Oct 2021

#LetsUnlitterUK: A demonstration and evaluation of the Behavior Change Wheel Methodology

PONE-D-21-24044R1

Dear Dr. Schmidtke,

We’re pleased to inform you that your manuscript has been judged scientifically suitable for publication and will be formally accepted for publication once it meets all outstanding technical requirements.

Kind regards,

Prof. Anat Gesser-Edelsburg, Ph.D.

Academic Editor

PLOS ONE

Additional Editor Comments (optional):

Reviewers' comments:

Reviewer's Responses to Questions

**Comments to the Author**

1. If the authors have adequately addressed your comments raised in a previous round of review and you feel that this manuscript is now acceptable for publication, you may indicate that here to bypass the “Comments to the Author” section, enter your conflict of interest statement in the “Confidential to Editor” section, and submit your "Accept" recommendation.

Reviewer #2: All comments have been addressed

2. Is the manuscript technically sound, and do the data support the conclusions?

Reviewer #2: Yes

3. Has the statistical analysis been performed appropriately and rigorously? 

Reviewer #2: Yes

4. Have the authors made all data underlying the findings in their manuscript fully available?

Reviewer #2: No

5. Is the manuscript presented in an intelligible fashion and written in standard English?

Reviewer #2: Yes

6. Review Comments to the Author

Reviewer #2: The authors have addressed all comments from my initial review. I have no further concerns at this time.

7. PLOS authors have the option to publish the peer review history of their article (what does this mean?). If published, this will include your full peer review and any attached files.

Reviewer #2: No

---

## [Editor Report · Acceptance letter]

4 Nov 2021

PONE-D-21-24044R1 

#LetsUnlitterUK: A demonstration and evaluation of the Behavior Change Wheel Methodology 

Dear Dr. Schmidtke:

I'm pleased to inform you that your manuscript has been deemed suitable for publication in PLOS ONE. Congratulations! Your manuscript is now with our production department. 

Kind regards, 

on behalf of

Prof. Anat Gesser-Edelsburg 

Academic Editor

PLOS ONE